# Finding an Appropriate Mouse Model to Study the Impact of a Treatment for Friedreich Ataxia on the Behavioral Phenotype

**DOI:** 10.3390/genes14081654

**Published:** 2023-08-19

**Authors:** Camille Bouchard, Catherine Gérard, Solange Gni-fiene Yanyabé, Nathalie Majeau, Malek Aloui, Gabrielle Buisson, Pouiré Yameogo, Vanessa Couture, Jacques P. Tremblay

**Affiliations:** 1Centre de Recherche du CHU, Québec-Université Laval, Québec, QC G1V 4G2, Canadanathalie.majeau@crchudequebec.ulaval.ca (N.M.); malek.aloui@crchudequebec.ulaval.ca (M.A.); gabrielle.buisson@crchudequebec.ulaval.ca (G.B.);; 2Département de Médecine Moléculaire, l’Université Laval Québec, Québec, QC G1V 4G2, Canada

**Keywords:** Friedreich ataxia, mouse model, phenotype, frataxin, shRNA

## Abstract

Friedreich ataxia (FRDA) is a progressive neurodegenerative disease caused by a GAA repeat in the intron 1 of the frataxin gene (FXN) leading to a lower expression of the frataxin protein. The YG8sR mice are Knock-Out (KO) for their murine frataxin gene but contain a human frataxin transgene derived from an FRDA patient with 300 GAA repeats. These mice are used as a FRDA model but even with a low frataxin concentration, their phenotype is mild. We aimed to find an optimized mouse model with a phenotype comparable to the human patients to study the impact of therapy on the phenotype. We compared two mouse models: the YG8sR injected with an AAV. PHP.B coding for a shRNA targeting the human frataxin gene and the YG8-800, a new mouse model with a human transgene containing 800 GAA repeats. Both mouse models were compared to Y47R mice containing nine GAA repeats that were considered healthy mice. Behavior tests (parallel rod floor apparatus, hanging test, inverted T beam, and notched beam test) were carried out from 2 to 11 months and significant differences were noticed for both YG8sR mice injected with an anti-FXN shRNA and the YG8-800 mice compared to healthy mice. In conclusion, YG8sR mice have a slight phenotype, and injecting them with an AAV-PHP.B expressing an shRNA targeting frataxin does increase their phenotype. The YG8-800 mice have a phenotype comparable to the human ataxic phenotype.

## 1. Introduction

Friedreich ataxia (FRDA) is the most common inherited recessive ataxia, a progressive and degenerative neurological disease. This ataxia is due to a mutation in the frataxin gene (FXN), which is, in most cases, an expansion in a GAA repeat in the first intron [1] leading to a reduced expression of the frataxin protein. The reduction of frataxin depends on the number of GAA repeats [2,3] and the protein concentration is drastically reduced in human patients [4]. This small protein is implicated in the iron transport and metabolism in mitochondria. Frataxin allows the acceleration of a sulfur transfer step, which limits the synthesis rate of 2Fe-2S clusters in the Fe-S assembly complex. Therefore, its decrease leads first to mitochondrial dysfunction and oxidative stress and finally to the death of cells including those in the cardiac and the nervous system [5,6]. Physical symptoms mean age of onset in classical FRDA is during childhood (6–16 years old) [7,8]. Patients suffering from Friedreich ataxia have difficulties with the coordination of their movements and their balance because of the loss of neurons in the central and peripheral nervous system. The premature death of patients is due principally to the cardiac damage associated with the disease [7,9,10].

Currently, there is no effective treatment for Friedreich ataxia. There are multiple potential therapies that are being explored, focusing on frataxin directly, such as the administration of frataxin protein fused with a cell-penetrating peptide [11], the induction of frataxin expression by the TALE protein fused with VP64 [12,13], and the administration of an AAV coding for the frataxin gene [14,15,16,17]. Another gene therapy approach is the excision of the GAA repeat with zinc finger nucleases (ZFNs) or with the CRISPR/Cas9 technologies [18,19,20]. These gene therapies are using an AAV to deliver the CRISPR system in mice. A Phase 1/2 clinical trial using an AAVrh10 to deliver the frataxin gene (LX2006) [21] to reduce FRDA cardiac symptoms has recently been approved by the Food and Drug Administration (FDA) [17].

Salami et al. [17] used a new mouse model, called αMyhc, created to reproduce the cardiomyopathy of the FRDA patients. The mouse model is of primary importance to obtain pre-clinical results. Different mouse models have been developed to investigate potential treatments, but so far, none of these models is close to the real phenotype of Friedreich ataxia. Puccio and coworkers in 2001 [22] developed the MCK-Cre cardiac mouse model expressing the Cre recombinase under the muscle creatine kinase (MCK) promoter permitting to knock out the FNX gene leading to cardiomyopathy after 5 weeks of age [23]. That group also developed the NSE-Cre nervous system mouse model expressing the Cre recombinase under the neuron-specific promoter enolase (NSE) [24]. The NSE-Cre model showed not only neurological deficits due to frataxin absence but also cardiac deficits leading to symptoms in both systems [22]. These mouse models were previously used in our laboratory, but the symptoms were too severe compared to the disease in humans. Indeed, their death occurred between 30 to 90 days for NSE-Cre and MCK-Cre respectively [15,25]. Recently, Puccio and co-workers generated a new conditional model called Pvalbtm1(Cre)Arbr/J using the parvalbumin promotor to target proprioceptive afferent sensory neurons such as the DRG, cerebellar Purkinje cells, and interneurons in the brain, to focus on the neuropathy associated with FRDA. These mice mimic the neuropathophysiology observed in FRDA patients but with a more rapid and severe course [16]. The KIKO model (Fxn(tm1Mkn/J) has an expansion of 230 GAA in one of the mouse Fxn genes and a deletion of the exon 4 in the other Fxn gene resulting in a robust neurobehavioral phenotype [26]. In 2017, Chandran and coworkers [27] created an inducible and reversible FRDAkd mouse model. This mouse model includes a transgene containing an shRNA against mouse frataxin under an H1 promotor, which was doxycycline-inducible. This permitted a significant frataxin protein reduction in all tissues. The strong decrease in mouse frataxin protein observed under doxycycline treatment was associated with a severe phenotype. The mice showed decreased locomotor activity and less coordination and were weaker and globally a phenotype, which is more similar to the Friedreich ataxia patients. However, this mouse model is inappropriate to test certain therapies targeting the human FXN gene such as deletion of the GAA repeats by CRISPR/Cas9, but the KIKO model could be used for this application.

Each mouse model has some advantages and some disadvantages, the goal is to choose the best one for the type of treatment to be investigated. The group of Pook [28] developed two Friedreich ataxia mouse models: the YG8R and the YG8sR in which both mouse frataxin genes are knockout and a human frataxin transgene derived from a Friedreich patient is added, which is advantageous for the investigations of some potential treatments. The original YG8R contained a human FXN transgene with two consecutive GAA repeats (one containing 90 and the other 190 GAA) while the Y47R control mouse had only 9 GAA repeats [28]. The improved YG8sR contained a human FXN transgene with only one repeat containing 250–300 GAAs [29]. However, even this YG8sR model expressed only a weak phenotype.

We, therefore, assumed that the weak phenotype in YG8sR mice was due to an insufficient reduction of frataxin. We decided to accentuate the severity of the YG8sR phenotype by using a shRNA against human frataxin mRNA, an approach similar to the one used by Chandran et al., for the mouse frataxin gene [27]. We initially selected an adequate shRNA by testing the effectiveness of different shRNAs to reduce frataxin in a culture of HeLa cells. The best shRNA was inserted in an AAV-PHP.B virus, which was injected intravenously, and the mice developed a more severe phenotype. The co-injection of two AAV-PHP.B codes for an anti-FXN shRNA and the frataxin gene used as a treatment prevented the development of a more severe FRDA phenotype.

In parallel, we compared this model to the new YG8-800 developed by Jackson Laboratory from the YG8sR descendants containing 800 GAA repeats instead of 300. The hypothesis here was that since a longer GAA expansion in humans correlates with a more severe phenotype [8], the same trend would be observed in mice.

## 2. Materials and Methods

### 2.1. Plasmids Construction (Coding for FXN Gene and Anti-FXN shRNA)

Adeno-associated viruses, serotype PHP.B (AAV-PHP.B) [30], were used to deliver either a plasmid coding for anti-FXN shRNA to increase the FRDA phenotype in YG8sR mice or the frataxin gene as a treatment in YG8sR mice. This specific serotype is known to drive expression in the central nervous system with an ability to cross the blood-brain barrier (BBB).

The previously described scAAV plasmid [15] was modified to change the CAG promoter for the chicken β-actin hybrid (CBh) promoter to express the human frataxin gene ubiquitously (scAAV-CBh-FXN). To synthetize the scAAV-hSyn-FXN, a specific plasmid to drive expression in the central nervous system, the scAAV-CBh-FXN plasmid was digested with AvrII and AgeI to remove the CBh promoter. The human synapsin 1 (hSyn) promoter was excised from plasmid AAV synap FZ (Addgene #60231) by XbaI and AgeI and ligated into the plasmid to create scAAV-hSyn-FXN. Both constructions were used to compare the efficiency of both promoters to deliver the FXN gene as a potential treatment.

For the construction of anti-FXN shRNA plasmids to increase the YG8sR phenotype, scAAV-hSyn-FXN was digested with BamHI and MluI, and U6-shRE and UBC-mCherry were ligated to the plasmid using NEBuilders. The resulting plasmid was digested with AgeI and NheI, and shFXN oligonucleotides were annealed and ligated to the plasmid to create scAAV-shRNAFXN (1, 3, 4 or 6)-UBC-mCherry plasmids. The shRNA sequences were derived from predesigned Sigma sequences: TRCN0000006138 (shRNA1), TRCN0000380594 (shRNA3), TRCN0000010996 (shRNA4), TRCN0000350511 (shRNA6). The sequences and sites targeted by the various shRNAs are illustrated in Figure 1A,B. The AAV plasmid containing the shRNA sequence is illustrated in Figure 1C.

### 2.2. In Vitro Testing of the shRNAs

HeLa cells were transfected using Lipofectamine 2000 with the different plasmids coding for a shRNA and mCherry (scAAV-shFXN-mCherry) alone or in combination for a total of 2 µg of plasmids. The fluorescence of mCherry was used to assess the transfection success. The cells were collected after 72 h and the proteins were extracted. The frataxin protein concentration was measured by an ELISA test (Ab109881, Abcam, Cambridge, UK). These in vitro tests allowed us to select the best shRNA likely to reduce frataxin in vivo.

### 2.3. AAV Production

For in vivo testing, different promotors were used to control the expression of the shRNA and the FXN gene. The shRNA was placed under a U6 promotor and the mCherry gene under a UBC (Figure 1C). In the AAV coding for the frataxin gene, two promoters were tested and compared, hSyn for specific delivery to the nervous system and CBh for a ubiquitous delivery.

Adeno-associated viruses, serotype PHP.B (AAV-PHP.B) [30], were used to deliver the shRNA or the frataxin gene in vivo in YG8sR mice. The AAV-PHP.B vectors were produced by the Plateforme d’outils moléculaires, Centre de recherche CERVO (Québec, QC, Canada).

### 2.4. AAV Injection

Mice (approximately 2 months old) were injected in the tail vein (IV) with 0.9% saline solution, with an AAV coding for shRNAscr (scramble shRNA sequence, not targeting frataxin) for control mice or with an AAV coding for anti-FXN shRNA. Different doses of AAVs were used from 0.5× (i.e., 3 × 10^11^ viral copies) to 4× (i.e., 2.4 × 10^12^ viral copies) depending on the effect on the mice. To treat the 2 months old mice, an AAV coding for human frataxin (FXN) gene was administrated alone or mixed with the AAV coding for a shRNA.

### 2.5. YG8sR, YG8-800 and Y47R Mice

The YG8sR, YG8-800, and Y47R mice were obtained from Jackson Laboratories Inc and reproduced in our facilities. To summarize, YG8sR (Fxntm1Mkn Tg(FXN)YG8Pook/2J; https://www.jax.org/strain/024113) (accessed on 10 June 2023) mice, YG8-800 (Fxnem2.1Lutzy Tg(FXN)YG8Pook/800J; https://www.jax.org/strain/030395) (accessed on 10 June 2023) mice and Y47R (Fxntm1Mkn Tg(FXN)Y47Pook/J; https://www.jax.org/strain/024097) (accessed on 10 June 2023) mice [29] all have mouse FXN genes knockout but they have, respectively, a transgene containing the human FXN gene with a 250–300 GAA triplet, an 800 GAA triplet, or a 9 GAA triplet. C57Bl6J (subsequently called C57) mice were used as control mice, containing no human transgene. The C57 mice group was made of 15 mice (9 females and 6 males), the Y47R group of 21 mice (13 females and 8 males), the YG8sR group of 19 mice (11 females and 8 males), and the YG8-800 group of 15 mice (7 males and 8 females) initially at 2 months age. The housing conditions were 2–5 mice per standard ventilated cage under a 12-h light/dark cycle with access to food and water ad libitum. Neslet and Aspen’s shaving are given for nidification. All experiments involving animals were approved by the Comité de Protection des Animaux de l’Université Laval (CPAUL3) (Québec, QC, Canada).

The first mice group injected with an AAV were euthanized at 3.5 months after the behavior tests, 5 weeks after the IV injection, and perfused with 0.9% saline to remove the blood from tissues. Different tissues (Tibialis anterior muscle, heart, liver, cerebrum, cerebellum, and dorsal root ganglions (DRG)) were then collected and stored at −80 °C. The subsequent group made of 15 C57 mice (9 females and 6 males), 21 Y47R mice (13 females and 8 males), 19 YG8sR mice (11 females and 8 males), and 15 YG8-800 mice (7 males and 8 females) were euthanized after 11 to 14 months with behavior tests every 3 weeks starting at 8 weeks age. The same tissues were also collected and stored at −80 °C. The euthanasia method was cardiac perfusion under isoflurane anesthesia.

### 2.6. Behavior Tests

The physical performances of YG8sR and YG8-800 mice were compared with those of Y47R mice and also with those of C57Bl6J (C57) mice, which are control mice with their natural mouse frataxin genes. All mice were evaluated every 3 months starting at 2 months and up to 11 months. For statistics, the females and males were pooled together because there were no significant differences in the preliminary analysis. The C57 mice group was made of 15 mice (9 females and 6 males), the Y47R group of 21 mice (13 females and 8 males), the YG8sR group of 19 mice (11 females and 8 males), and the YG8-800 group of 15 mice (7 males and 8 females).

#### 2.6.1. Parallel Rod Floor

The mice were placed in the parallel rod floor apparatus [31] for 8 min and followed with a camera. The apparatus is a plexiglass cage with metallic parallel rods 1 cm above the metallic plate floor. The apparatus detected foot faults when the foot passed through the grid and touched the metallic plate at the bottom. Different parameters (i.e., the distance traveled, the average speed, the time spent immobile, and the foot fault number) were recorded with the ANY-maze software version 7.20 connected to the camera and floor foot detector.

#### 2.6.2. Hanging Test

The mice were weighed before the test. This is a four-limbs hanging test, the mouse grasped a wire grid as it is inverted. The test was based on the Kondziela’s inverted screen test [32,33]. The time of sustained limb tension to oppose the mouse weight was measured. The chronometer was started as soon as the grid was inverted over the cage. The time the mouse remained hanging (before the mouse fell) was noted, with a maximum of 300 s for the test. Three sets of hanging tests were conducted with a minimum of 2 min between each test. The longest time the mouse remained hanging on the screen was noted.

#### 2.6.3. Notched Beam Test

Our test was based on the Di Bonito [34] notched beam test. We used a one-meter Plexiglass beam engraved with 18 squares of 2 cm wide/deep/long and spaced every 2 cm. The locomotion was assessed by the time required for the mouse to cross the beam over the squares. We also counted the foot faults manually with 2 observers (i.e., each time the foot touched the bottom of the square). The experiment was made in triplicates and the average number of foot faults was used.

#### 2.6.4. Inverted T Beam/Balance Beam Test

This test used a Plexiglass balance beam, which was 1 m long and 2 cm wide with a central platform of 0.66 cm wide and 0.66 cm high based on the balance beam test of Luong and co-workers [35]. The mouse walks on the 0.66 cm top beam and steps on the bottom parts of the platform (inverted T shape) when making a foot fault. The time to cross the beam was also recorded as well as every foot fault (mouse foot slipped off the beam), which were manually counted by 2 people. The experiment was made in triplicates and the average number of foot faults was used.

### 2.7. Imaging of mCherry Fluorescence in Mouse Organs

The integrated imaging station IVIS Lumina LT Series III (PerkinElmer, Waltham, USA) was used to evaluate the distribution of mCherry protein in macroscopic organs using its intrinsic orange-red fluorescence. The apparatus was set in fluorescence mode with filters at 535 nm (excitation) and 620 nm (emission), small binning, and 1–30 s range of exposure time. Under the dedicated Living Image software 4.5.4, the fluorescent signal intensity was calibrated as radiant efficiency and normalized to allow intensity comparisons. No significant auto-fluorescent signal was detected. The imaging was achieved by the Plateforme Imagerie animale par bioluminescence/fluorescence, Centre de recherche CHU (Québec, QC, Canada).

### 2.8. DNA Extraction from Tissues

DNA was extracted from different tissues (muscle, liver, heart, brain, and dorsal root ganglions). Briefly, a part of the tissue was recovered and incubated with 50 mL proteinase K (10 mg/mL) in a lysis buffer (1.82 g TrisBase, 5 g SDS, 10 mL EDTA 0.5 M completed with distilled water to 500 mL, pH 8.0) at 56 °C until the solution became clear. Digested tissues were then mixed with a 500 µL solution of phenol/chloroform/isoamyl alcohol (25:24:1; BioShop Canada Inc., Burlington, ON, Canada) and centrifuged for 3 min at 12,000 rpm. The upper solution was recovered and mixed with the same volume of chloroform and centrifuged again. The upper solution was recovered, and 50 µL of 5 M sodium chloride was added before the addition of 1 mL 100% ethanol. After centrifugation for 8 min at 12,000 rpm, the pellets were washed in 70% alcohol before another centrifugation. Pellets were dried before DNA suspension in sterile water. The detection of the AAV content was accomplished by PCR, with a band of 410 pb and 435 pb respectively for the plasmid delivered by AAV expressing the shRNA (AAV-shRNA) or the frataxin gene (AAV-FXN) (Appendix A). The shRNA plasmid contained a 410 pb mCherry section (Figure 1C) and the frataxin gene DNA amplified fragment was 435 pb.

### 2.9. Quantification of the Viral Particles by qPCR

The viral particle concentration in organs was measured by qPCR using primers that target the mCherry sequence present with the shRNA sequence in the AAV. 50 ng of the previously extracted gDNA was used per well with 6 µL Advanced Universal SYBR Green supermix, 0.5 µL 10 mM Forward mCherry primer, 0.5 µL 10 mM Reverse mCherry primer and completed to 10 µL with DNase free water. Bio-Rad CFX96 was set at SYBR only with a 3-min 98 °C polymerase activation and DNA denaturation phase, then 40 cycles of 10 s 98 °C denaturation and 20 s 60 °C annealing and extension with a melt curve analysis from 65 °C to 95 °C increasing the temperature by 0.5 °C at every 5 s. The standard curve was made with 8 samples from 101 to 108 particles. Quantified organs were TA muscle, liver, heart, cerebrum, cerebellum, and DRG. All samples were duplicated. This qPCR used primers in the mCherry and in the HPRT reference gene (Appendix A).

### 2.10. Quantification of mRNA Expression by qRT-PCR

Frataxin mRNA expression in tissues was compared by qRT-PCR. The RNA was first extracted by adding 1 mL of Trizol to the finely crushed tissue slice. A total of 200 µL chloroform was then added to separate the phases during 15 min centrifugation at 12,000 rpm. The supernatant was collected and precipitated in 500 µL cold (−20 °C) isopropanol centrifuged at 12,000 rpm for 10 min. The liquid was then removed from the tube to keep only the pellet to be washed twice with 1 mL 75% cold (−20 °C) ethanol and centrifuged at 10,000 rpm for 5 min. The pellet was resuspended in 20–60 µL RNAse free water. The RNA concentration was measured with a spectrophotometer and the migration on agarose gel 1% permitted to verify that the RNA was not degraded. The sample was then treated with DNAse (NEB Inc.) incubated for 10 min at 37 °C, then the DNAse was deactivated with EDTA 2.5 mM at 75 °C for 10 min. The RNA was then put in the presence of Reverse Transcriptase with dNTPs and random primers in the buffer provided by Thermo Fischer Scientific Inc. (Waltham, MA, USA) in the ‘High Capacity cDNA Reverse Transcription kit’. A qPCR was then performed with the complementary DNA obtained. This qRT-PCR used primers for frataxin as well as for HPRT, a reference gene (Appendix A).

### 2.11. Frataxin Protein Quantification

The proteins were extracted using 300 µL of the protein extraction buffer provided by AbCam in the Human frataxin Elisa kit (ab176112, Abcam, Cambridge, MA, USA) per tissue slice (0.5 mm). The frataxin protein concentration was estimated using the PierceTM BCA Protein Assay Kit (Thermo Fischer Scientific Inc.). This kit uses a bovine serum albumin (BSA) standard of 0 mg/mL, 0.25 mg/mL, 0.625 mg/mL, 1.25 mg/mL, 1.875 mg/mL, 2.5 mg/mL and 5 mg/mL. All samples were conducted in duplicate and their optical density (OD) at 562 nm was compared to the standard curve to estimate their protein concentration.

The human frataxin protein was quantified using the Human frataxin Elisa kit (ab176112, Abcam, Cambridge, MA, USA) for in vitro and for mouse tissues. A standard curve was also made at the same time with recombinant human frataxin protein to quantify the frataxin protein in the tissues in nanogram/microgram total protein. All samples were carried out in duplicate and compared to the standard curve using their OD at 450 nm.

### 2.12. Statistical Analysis

The results are presented as mean ± SD. Statistical analyses were performed using GraphPad Prism7 software version 9.0 (GraphPad Inc., La Jolla, CA, USA) and detailed in each figure legend. The results were analyzed in two ways ANOVA test Tukey’s multiple comparisons test (* *p* value ˂ 0.05, ** *p* value ˂ 0.005, *** *p* value ˂ 0.0002 and **** ˂ 0.0001) for behavior tests. For qPCR the statistics were conducted by the software for the qPCR Biorad CFX Maestro. The *p* values are indicated in each figure. Statistics: one-way ANOVA was used at 0.05 (95% confidence interval); * *p* < 0.05, ** *p* < 0.003, *** *p* < 0.0003, and **** *p* < 0.0001.

## 3. Results

### 3.1. In Vivo Comparison of YG8sR, YG8-800, Y47R and C57BL6J Mice

The YG8sR mouse is a model of Friedreich ataxia, having both mouse frataxin genes knockout but containing a human frataxin transgene with 250–300 GAA repeats. The YG8-800 mice are descendants from YG8sR but contain 800 GAA repeats. The control mouse for this model is the Y47R mouse, which also has both mouse frataxin genes knockout but has a human frataxin transgene with only 9 GAA repeats.

#### 3.1.1. Behavior Tests

The physical performances of YG8sR and YG8-800 mice were compared to Y47R and C57Bl6J (C57) control mice.

The body weight (Figure 2A) was measured at each test session. The YG8sR mice had a significantly higher body weight than the Y47R mice starting at 5 months. The YG8-800 had a significantly lower weight than both control mice at 5 and 8 months. For the hanging test, where the mouse grips on an inverted grid (Figure 2B), There was a significant decrease in time hanging per body weight for YG8sR mice compared with the Y47R control mice at 5 and 8 months and also for the YG8-800 mice compared with the Y47R at 8 months.

The parallel rod floor is an apparatus made of an enclosure box with a grid on the floor and a camera above recording the distance traveled by the mouse, the average speed, the time immobile, and the foot faults during 8 min. The distance traveled by all mice decreased slowly from 2 to 11 months (Figure 2C). The distance is also significantly inferior for YG8-800 mice compared to both controls (C57 and Y47R) at 5 and 8 months and for YG8sR mice compared to the C57 model also at 5 and 8 months. The average speed (Figure 2D) was also significantly lower for the YG8-800 mice compared to both control models at 5 and 8 months. The difference was significantly inferior for the YG8sR compared to the C57 control at 5 months. The time spent immobile (Figure 2E) for YG8-800 mice was significantly higher than both controls at 5 months. At 8 months, it is the YG8sR mice that become significantly less active than the Y47R. The amount of foot faults per distance (in m) (Figure 2F) is significantly higher for both YG8sR and YG8-800 compared to the Y47R control at 8 and 11 months.

Two types of beams were also used to test the behavior: the notched beam where the mice walk from the top of a block to another block and the inverted T beam where the mice walk on a narrow surface. For the notched beam, there is a significant increase in the time needed to cross the beam for YG8-800 compared to C57 at all times and also for the YG8sR compared to C57 at 11 months (Figure 2G). The foot faults (Figure 2H) significantly increased for YG8-800 mice compared to both C57 and Y47R controls at 5, 8, and 11 months. A significant foot fault increase for YG8sR mice is also seen at 11 months compared to C57 mice. The time spent on the inverted T beam (Figure 2I) is only significantly higher for YG8-800 mice compared to YG8sR at 2 months but was not significantly different for all mice at 5, 8, and 11 months. The number of foot faults on the inverted T beam (Figure 2J) was significantly superior for YG8-800 mice compared to both controls at 2, 5, and 8 months. At 11 months, the YG8sR mice made significantly more foot faults than the C57 control mice.

To summarize, the physical performance of C57, Y47R, and YG8sR mice from 2 to 11 months old were similar for many tests. The YG8sR mice showed a mild movement coordination impairment phenotype and the YG8-800 mice, a more severe one.

#### 3.1.2. Frataxin Concentration in Mouse Organs

The human frataxin protein concentration was measured in several tissues, i.e., the Tibialis anterior muscle, the liver, the heart, the cerebellum, the cerebrum, and dorsal root ganglions (Figure 3A). Even if there were great variations between mice, the frataxin concentration was significantly lower in the YG8sR mice compared with the Y47R mice. Indeed, the frataxin concentration in the YG8sR mice was only 13% to 20% of the frataxin concentration in the Y47R mice, depending on the tissues (Figure 3B).

Our aim was to identify a mouse model with clear ataxic phenotypes, which was not the case for the YG8sR mice. Even if the frataxin concentration in tissues is low, we thus decided to further decrease the frataxin concentration with an shRNA against human frataxin (the YG8sR mice have a human frataxin transgene) to trigger a phenotype earlier, which might be closer to the human pathology.

We also added the YG8-800 mouse model to our study to evaluate its phenotype, since it had not been characterized yet. The decrease in human frataxin is more severe in YG8-800 mice than in the YG8sR mice. When compared to Y47R mice, YG8-800 mice have between 0.9% and 16.3% of the Y47R human frataxin concentration in tissues (Figure 3B). This is therefore the aimed concentration for YG8sR mice injected with a shRNA targeting the human frataxin gene since this frataxin level induces an ataxic phenotype which is comparable to the human patients.

### 3.2. In Vitro and in Vivo Testing of Different shRNAs against Frataxin

#### 3.2.1. In Vitro Testing of 4 shRNAs against Frataxin

The sequence of one of four shRNAs (Figure 1A) against the human frataxin mRNA or a control scramble shRNA (shRNAscr) was inserted under the control of a U6 promoter in an AAV-PHP.B vector. The plasmids coding for these shRNAs also contained a mCherry gene under a UBC promoter to monitor the transfection efficiency by fluorescence (Figure 1C). The different plasmids containing the shRNA were transfected in HeLa cells. After 72 h of incubation, the cells were recovered, and proteins were extracted to quantify the frataxin concentration. The transfection of the control shRNAscr did not affect the frataxin concentration in the HeLa cells compared to the untransfected cells, which confirmed that the transfection of the plasmids by themselves did not affect human frataxin expression (Figure 4). In contrast, each shRNA targeting the frataxin mRNA led to a significant decrease in frataxin concentration in the transfected cells. The shRNA6 decreased the frataxin concentration close to 50% of controls and the shRNA4 to about 40%. The shRNA1 and the shRNA3 reduced frataxin concentration the most, to a concentration of 28.8% and 28.3% respectively of the control cells frataxin. Combinations of two shRNAs were also tested in HeLa cells to evaluate if their association could decrease even more the frataxin concentration. All combinations also significantly reduced the frataxin protein concentration compared to the control but none of them decreased the frataxin significantly more than one shRNA alone.

Following these results, we selected the three shRNAs, two that were more effective for decreasing the frataxin concentration, i.e., shRNA1 and shRNA3, and also shRNA6 although it was less effective. We also kept the shRNAscr as a negative control. These shRNAs were also integrated into an AAV-PHP.B virus to be tested in vivo. Error bars indicate standard deviation (SD).

#### 3.2.2. In Vivo Tests of the shRNAs against Frataxin

##### The Impact of Anti-FXN shRNAs on the Coordination of YG8sR Mice

To increase the severity of the impaired coordination phenotype, we administrated intravenously in YG8sR mice an AAV-PHP.B coding for one anti-FXN shRNA previously validated in vitro or for the negative control shRNAscr. Different concentrations of AAVs coding for an shRNA were tested in vivo to obtain a progressive worsening phenotype, which would be severe enough to observe a loss of coordination 5 weeks after injection. The AAV concentrations chosen were 2× and 4× (respectively 12 and 24 × 10^11^ viral particles) for shRNA1 and shRNA6 and 0.5× and 1× (respectively 3 and 6 × 10^11^ viral particles) for shRNA3 (Appendix A). The concentration of shRNA3 injected was lower than for shRNA1 because, in preliminary experiments, the mice injected with AAV shRNA3 had to be sacrificed earlier, around 15 days after the AAV injection at 2× due to the severity of the mouse illness. For the shRNAscr, three concentrations were tested 6, 12, and 24 × 10^11^ viral particles.

Different behavior tests were made before the AAV injection as training for the mice at 5 weeks after injection. At 5 weeks after injection, no significant differences were observed in the parallel rod floor (results not shown). Two beam tests were used, the notched beam test (Appendix A) where mice crossed the beam on the top of blocks, and the inverted T beam (walk beam) test (Appendix A), where the mice tried to cross the beam on a thin top. The mice were also tested for their muscle strength with the hanging test. Based on all these tests, the shRNA3 was determined as the best shRNA to induce an ataxic phenotype (Appendix A). We, therefore, focused on the shRNA3 for the following studies.

The results obtained with the mice injected with the AAV-PHP.B-shRNAscr are similar to those obtained with the untreated mice, i.e., saline-injected mice (Figure 5). The two doses of AAV shRNA3 led to a more severe disease phenotype by increasing the time to cross the two beams and increasing the number of foot faults (Figure 5A–D) but also by decreasing the time the mice stayed hanging on the grid (Figure 5E).

The best shRNA was the shRNA3 to induce not only a loss of coordination observed on the beams but also a loss of muscle strength determined with the hanging wire test (Figure 5E) and that, at a lower concentration of virus than the shRNA1 administration.

##### Virus Detection

At the time of sacrifice, different organs were removed: the Tibialis anterior muscle (M), the cerebrum (Ce), the cerebellum (Cr), the liver (L), the heart (H), and the spine containing the dorsal root ganglions (DRG). All the organs were placed into the IVIS Lumina LT Series III and excited to detect the fluorescence of the mCherry protein (Appendix A). Fluorescence was observed in the tissues of the mice treated with all the shRNAs against frataxin or with the shRNAscr but not in the tissues of the control mice injected with saline (Appendix A right side). The fluorescence was most intense in the brain followed by the liver, the heart, and the spine. However, the low fluorescence of the spine may be due to the fact that the bones and tissues surrounding it were not removed. No fluorescence was detected in the muscles. Since these data provide indirect evidence of shRNA concentration, they are available in Appendix A.

The genomic DNA was extracted from each tissue and a PCR was made to detect the expression of the mCherry gene contained in the AAV. The AAV-PHP.B serotype was used, which is known to cross the blood-brain barrier and concentrate in the nervous system. A positive control was made with the direct plasmid to confirm the band of mCherry at 410 pb. A band was observed in each tissue of mice injected with each shRNA and the shRNAscr (Appendix A). The band was more intense in the nervous tissues (cerebrum, cerebellum, and DRG) and the band was the lightest in the muscles of the injected mice. No band was observed in the control mice injected with saline (Appendix A).

A q-PCR was also conducted to estimate the number of viral particles in each tissue, also by detecting the mCherry expression (Appendix A). The number of viral particles was the highest in the DRG, while the cerebrum and cerebellum had similar but slightly lower concentrations of AAV. Lower concentrations of AAVs were progressively detected in the liver, the heart, and muscle of the injected mice.

#### 3.2.3. Quantification of the Frataxin Expression and Concentration

The frataxin mRNA concentration was evaluated in three organs, i.e., the liver, as an organ of detoxification, the cerebrum, and the cerebellum for the nervous system. RNA was extracted from these tissues and a qRT-PCR was made to quantify the expression of human frataxin in these tissues. The proteins of the tissues were also extracted and the human frataxin concentration was quantified and expressed as a percentage reported to the saline-treated mice. The concentrations of the frataxin mRNA and the frataxin protein were similar for the mice injected with saline or with the scramble shRNA, except for the liver (Figure 6A,B). In the liver, both frataxin mRNA and frataxin protein concentrations were significantly reduced by 37.7% and 50% respectively in the mice injected with the shRNA3 (Figure 6A,B). The shRNAscr induced a significant increase of the frataxin mRNA compared to the saline control in the liver, thus if compared to the shRNAscr and the shRNA3, a significant reduction of close to 60% was measured with the shRNA3. The frataxin mRNA expression was also significantly reduced by 29.3% in the cerebellum but no difference was observed for the protein with the control mice (Figure 6C,D). No significant differences were observed in the cerebrum neither for the mRNA expression nor for the frataxin concentration (Figure 6E,F).

#### 3.2.4. Comparison between Two Promoters for the Treatment with an AAV Coding for Human Frataxin

With the intent to develop a treatment by delivering the human frataxin gene systemically in mice, a part of this project was to use AAV.PHP.B with neuro-specific and non-neuro-specific promoters to deliver the human frataxin gene and study the impact on the movement coordination. The AAV.PHP.B allows to cross the BBB but is not neuro-specific. Therefore, AAV.PHP.B expressing human frataxin under two different promoters were tested, i.e., CBh-FXN for a more general expression and hSyn-FXN for an expression focused on the nervous system. The recombinant FXN gene present in the AAV has an almost identical nucleotide sequence to the human FXN gene expressed in the mouse but with some differences in the nucleotide sequence so that the shRNA3 cannot reduce the human FXN expression induced by the virus. These two AAVs coding for the human FXN were tested alone or in combination with AAVs coding for the shRNAscr or the shRNA3. Because all AAVs were of the same serotype (i.e., AAV-PHP.B), the treatment (CBh-FXN or hSyn-FXN) and the shRNA were injected at the same time, and mouse behavior tests were made 5 weeks after the AAV injection (Figure 7). In these behavior tests, the treatment with an AAV coding for frataxin under the CBh or the hSyn promoter alone or in association with the shRNAscr did not affect the behavior of the YG8sR mice for both beam tests (Figure 7). In the notched beam, both treatments (i.e., FNX under the CBh or the hSyn promoter) co-injected with the shRNA3 counterbalanced the shRNA inhibitory effect. Indeed, the time and the number of foot faults made by crossing the notched beam of the co-injected mice were similar to the performance of the saline-treated mice (Figure 7A,B). In the inverted T beam, although the treated mice crossed the beam as rapidly as the controls, the number of foot faults stayed similar to the saline control mice only with the injection of CBh-FXN (Figure 7C,D). Figure 7E represents the results of the hanging wire test. Both treatments, CBh-FXN and hSyn-FXN, restored the ability of the mice to remain gripped for an extended period of time. However, when examining the individual mouse performance for the hSyn-FXN treatment, 3 out of 4 mice fell very fast, and only one stayed longer. On the contrary, with the CBh-FXN treatment, all mice stayed on the grid much longer. The treatment with the CBh-FXN alone or with shRNA co-injection even increased the time that the mice stayed hanging compared to the saline control mice.

Before analysis of the tissues, genomic DNA was extracted, and PCRs were made to detect the mCherry gene present in the shRNA vector and the human FXN gene in mice that received the AAV coding for that gene to verify that each mouse was well injected with the right treatment (Appendix A).

The expression of human frataxin mRNA was also quantified 5 weeks after the IV injection of the AAV and reported as fold increases compared to saline-treated mice (Table 1). In the cerebrum and cerebellum, the frataxin increase was comparable between CBh-FXN and hSyn-FXN alone or in co-injection with the shRNA3. On the contrary, in the liver, the concentration of frataxin was greatly increased with the CBh promoter comparatively to the hSyn promoter which does not significantly affect the expression and concentration of frataxin.

## 4. Discussion

Initially, Virmouni and co-workers developed a new mouse model containing a human FNX transgene with about 200 GAA repeats, here called YG8sR(200) for an easier discussion [29]. Even if the behavior of these YG8sR(200) mice on the rotarod was significantly different from C57 and Y47R mice, by pooling males and females together, the differences were not as clear as the results obtained when the sexes were analyzed separately. Moreover, the YG8sR(200) mice showed reduced locomotion activity over time but the most significant differences were observed with more specific tests such as the beam tests, the hang-wire test, and the grip test at 12 months. Indeed, at that age, the YG8sR(200) mice took more time to cross the two beams, fell faster on the hang-wire test, and showed less grip strength compared to the Y47R or C57 mice. The human frataxin mRNA, as well as the human frataxin protein concentrations in YG8sR(200) mice, were also decreased in the brain, the cerebellum, and the liver at 12 months compared to Y47R and C57 mice [29]. We have purchased the YG8sR mice from Jackson Laboratories Inc. and those that we have used contained between 250 and 300 GAA repeats (https://www.jax.org/strain/024113) (accessed on 10 June 2023). In our studies, the YG8sR females performed better than the YG8sR males in the behavior tests. Nevertheless, no great difference was observed compared to Y47R mice in these tests. The concentration of human frataxin protein was reduced in YG8sR in all tissues tested compared to Y47R and even more than in the YG8sR(200) for the cerebellum (i.e., 70% in the YG8sR(200) and 19% in YG8sR). Considering that YG8sR mice have a human frataxin transgene containing 250–300 GAA repeats responsible for low human frataxin expression, these mice remain a good model to study the effects of frataxin reduction [36], or to try to increase the frataxin expression by some treatments [37,38] or even through genetic modifications using the CRISPR-Cas9 system to delete the GAA repeat [19,20]. All these studies are essential to understanding the mechanism and interaction between the frataxin protein and other proteins or enzymes but at some point, it becomes essential to have a good mouse model that recreates the symptoms of Friedreich ataxia and allows for the detection of functional improvements with treatment.

The use of shRNAs is a method used to decrease the expression of a protein and can be tissue-specific by using a specific promoter. For example, by using a member of the cadherin family that is exclusively expressed in the renal tubular epithelial cells, i.e., Ksp-cadherin, Xu et al. [39] observed 2 weeks after IV injection of an AAV9-shRNA-ALDH2, a decrease of the aldehyde dehydrogenase 2 protein (ALDH2) only in the kidney although this protein is also expressed in the liver and the heart [39]. Chandran and co-workers in 2017 [27] used shRNAs to decrease the mouse frataxin and create a mouse model for FRDA called FRDAkd. In their study, different shRNAs against frataxin were tested and they decided to use one, which induced the strongest reduction of frataxin in vitro. Among the different shRNAs that they tested, the one decreasing the most the frataxin expression was also targeting exon 4, as in our case for our sh1 and sh3 [27].

Piras et al., in 2013 suggested that gene knockdown is generally less predictable than gene over-expression [40]. In our study, the concentration of viral particles in the brain (cerebrum and cerebellum) was as high as in the liver but the frataxin concentration was significantly reduced only in the liver. In our case, the AAV serotype PHP.B was used. In the first study on AAV-PHP.B, the bio-distribution after IV injection in mice was compared to AAV9 and was similar between the two AAVs in the liver, the heart, and the skeletal muscle but the expression of GFP was lower with the AAV-PHP.B than with AAV9, even in the liver, where viral particles were as high as in the nervous system [30].

Chandran and co-workers also created a new mouse model for Friedreich Ataxia. They incorporated in a defined genomic locus, a single copy of a doxycycline-inducible anti-FXN shRNA to silence frataxin. Different behavior tests were carried out at 12 and 24 weeks of age with doxycycline treatment and they observed neurological deficits. After 20 weeks of age, the mice showed a drastic reduction of frataxin protein in the eight organs tested (liver, heart, kidney, brain, muscle, pancreas, lung and spinal cord) [27]. In our experiments, the effect of shRNA was faster and the frataxin mRNAs and protein concentrations were estimated only 5 weeks after the AAV-shRNA injection. However, after 5 weeks, a reduction of frataxin protein was observed in the liver but not in the other organs. Nevertheless, the frataxin mRNA was significantly reduced in the liver but also in the cerebellum. It is probable that 5 weeks is too short to observe an effect on the frataxin concentration on other organs than the liver because in the Chandran et al. study, the investigators showed a decrease of the frataxin protein concentration only for the liver in a time series starting at 3 weeks of age, but in other organs after 20 weeks [27].

The important question is whether we can reverse a phenotype even in patients who have lived several years with the disease by increasing frataxin expression. Many researchers tried to answer this question. Chandran et al. [27] with their inducible mouse model FRDAkd were able to stop the treatment with the doxycycline to increase again the expression of frataxin. After 13 weeks of doxycycline treatment followed by 12 weeks without this treatment, the mice recovered a healthy phenotype close to control mice [27]. Therefore, it appears that the coordination impairment is reversible when frataxin is increased. Another method to increase the concentration of frataxin in mice is to inject an AAV coding for the frataxin gene. We have worked on the delivery of the frataxin gene by AAVs since 2014 [15]. We initially used an AAV9 for the delivery in NSE-cre and MCK-cre mice, i.e., the mouse models created by Puccio H. in 2001 [22]. We found a great improvement in heart function and behavior tests by intraperitoneal administration of AAV9-FXN at an early age, between 5 and 9 days of age [15]. In the present study, our goal was to focus on the neuronal dysfunction, therefore, the AAV-PHP.B was chosen to target the nervous system with the shRNA. A co-injection (AAV-shRNA and AAV-FXN) was necessary to avoid a hyperacute immune response that could be provoked by a second AAV injection [41]. In our study, the co-injection of AAV-PHP.B-CBh-FXN with an AAV-PHP.B-antiFXNshRNA improved the behavior of mice compared to those treated with the AAV-PHP.B shRNA against frataxin alone. Puccio and co-workers created in 2018 a mouse model to specifically target cells expressing parvalbumin, called parvalbumin conditional knockout mouse model (Pvalb cKO mouse) [16]. The parvalbumin promoter is specific to proprioceptive neurons. These mice reflected the FRDA neuropathophysiology. They injected intravenously into 3.5 weeks old mice an AAV9-CAG-FXN and obtained a significant coordination improvement in the notched beam but not reaching the phenotype of their control mice (Fxn+/L3). Moreover, in the wire-hanging test, the treated Pvalb cKO mice were similar to the WT. Our aim was to have a comparable coordination impairment to this model in a mouse model containing the human gene with an expanded GAA to evaluate the impact of a genetic treatment on the coordination of mice. We used AAV. PHP.B to cross the BBB, but tissue analysis revealed not only the efficiency of the AAV9 on the sensory neuropathy but also a non-homogenous distribution of the AAVrh10 in the cerebellum leading to a partial rescue of Purkinje cells [16]. AAV8 was also tested in the FRDAkd mouse model (described above) in 2022 [42]. Ten weeks after dox treatment the mice received an intravenous injection of AAV8 coding for frataxin. Unfortunately, restoration of frataxin with this AAV failed to improve the behavior, metabolic and histological features of this mouse model [42].

The different articles using an AAV coding for frataxin were often using a strong promoter such as CMV or CBh which led to overexpression of frataxin in different organs. A recent review by Sivakumar and Cherqui in 2022 highlighted the controversial effects of frataxin overexpression [43]. In the past, in our studies, even if we obtained overexpression of frataxin in the tissues of the mouse models (and especially in the liver), we never found clear evidence that overexpression was harmful to the mice. This may depend on the frataxin concentration which was obtained. Indeed, Puccio’s group published in 2020 [44] a study showing that a dose-dependent harmful effect was observed in mouse heart function started after a 20-fold increase in frataxin concentration. However, the harmful effect was tissue-specific since a 90-fold increase of frataxin in the liver of mice, did not induce toxicity. In the present study, we tested two promoters, a general and strong (CBh) one and a weak one more specific for the nervous system (hSyn). The frataxin expression under a CBh promoter showed at 5 weeks post injection a significant improvement in the behavior tests compared to the hSyn promoter. Nevertheless, the period of testing was just 5 weeks post-injection and it is possible that this follow-up period was too short to observe an improvement in the behavior with the hSyn promoter. Moreover, the hSyn promoter, due to its specificity, did not induce any increase of frataxin in the liver even if the number of viral particles was high.

It is important to find an appropriate mouse model of FRDA for what is being tested, e.g., to measure the impact of treatment on the behavioral phenotype. Mercado-Ayón et al., (2022) [42], demonstrated that the symptoms obtained in their FRDAkd mouse model are not only the result of the reduction of frataxin expression but that the doxorubicin treatment has also an impact on the symptoms. In our study, the shRNA was provided by an AAV and no additional treatment was necessary. The only disadvantage of our approach is that we had to co-inject the treatment and the shRNA because the same AAV serotype was used for both and thus a delayed administration of the AAV was not possible because of the immune response induced by the first administration.

The aim of this study was to decrease the frataxin in YG8sR mice to increase the severity of their phenotype. Therefore, both YG8sR injected with an shRNA and YG8-800 mice phenotypes were compared. The results have shown that shRNA targeting frataxin does decrease its expression and increases the ataxic phenotype of YG8sR mice. The question was if this method is now the optimal one knowing that the YG8-800 mouse model is available. The complete characterization of this model has been carried out in our laboratory [45] showing that the phenotype of the YG8-800 mice is comparable to the human patients since the movement coordination progressively decreases and a cardiac hypertrophy is noticed. When compared, the YG8sR mice injected with the shRNA3 (Figure 5) have a phenotype comparable to the YG8-800 mice (Figure 2G–J). In fact, at 3–4 months, the YG8sR +shRNA3 need an average of 35 s to cross the notched beam and make 12-foot faults, which is slightly more severe than the YG8-800 mice 20 s and 7-foot faults at 5 months (saline control needing in average 10 s and making around 2.5-foot faults). The same trend is noticed for the inverted T beam, on which the shRNA-treated mice need on average 57 s and make 19-foot faults, whereas YG8-800 mice need 18 s and make 14-foot faults (saline needs 9 s and make 6-foot faults). In conclusion, both YG8sR injected with shRNA3 and YG8-800 offer a phenotype that can be compared to human patients. The YG8-800 mouse model is recent and has only a few articles about it. In September 2022, our team published the characterization of this mouse model including the cardiac hypertrophy which is also part of the human phenotype [45]. In January 2023, Kalef-Ezra et al. confirmed the impaired coordination and reduced frataxin levels of the mouse model [46].

We also studied the possibility of using AAV.PHP.B to deliver the frataxin gene. Our results on YG8sR mice were promising since they reestablished the coordination and more tests will be conducted in YG8-800 mice without needing to co-inject with an anti-FXN shRNA.

## 5. Conclusions

In our study, we were able to rapidly increase the severity of the YG8sR phenotype with an AAV coding for a sh-RNA against frataxin. Indeed, only 5 weeks after this AAV injection, the mice demonstrated significantly more severe locomotion and balance symptoms. A co-injection of an AAV coding for the same shRNA and of an AAV (same serotype) coding for frataxin prevented the development of disease symptoms. AAV-based gene therapy remains for the moment the best method to deliver a DNA sequence and the development of new serotypes opens the possibility of developing more tissue-specific delivery.

## Figures and Tables

**Figure 1 genes-14-01654-f001:**
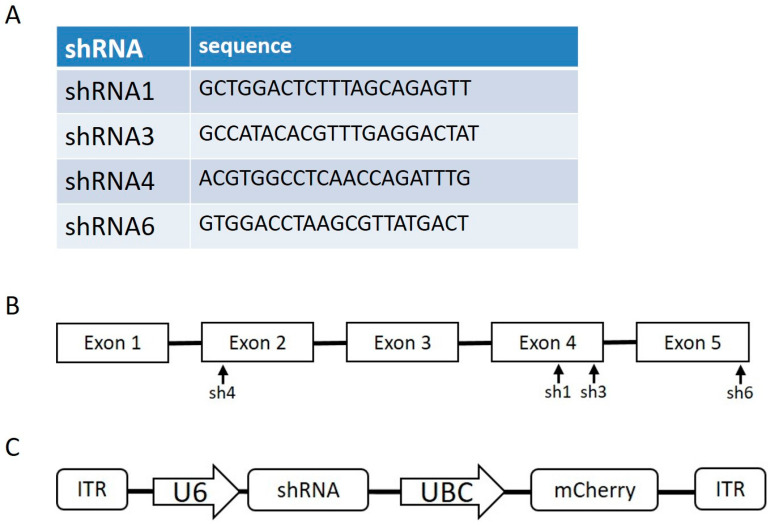
Specific sequences in the frataxin gene targeted by 4 shRNAs are indicated in (**A**). The sites of these 4 shRNAs are indicated in (**B**). The shRNAs were integrated into AAV vectors under the U6 promoter and these AAVs also contained a mCherry gene under the UBC promoter (**C**).

**Figure 2 genes-14-01654-f002:**
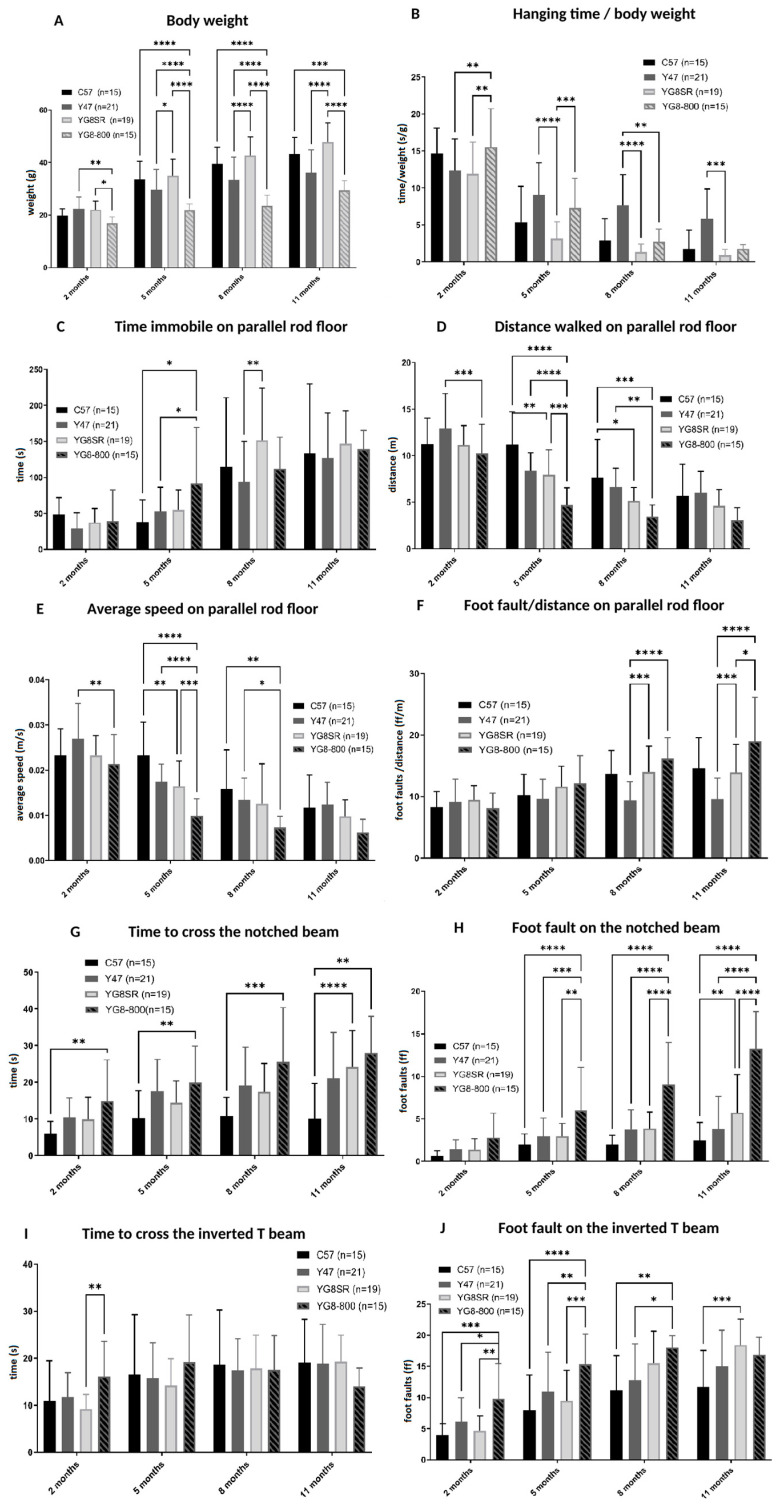
The C57, Y47R, YG8sR, and YG8-800 mice were followed with different behavior tests for 11 months. The body weight was determined (**A**) and served for the hanging wire test (**B**). The parallel rod floor is a cage with a wire rack on the floor and the following parameters were analyzed during 8 min: the distance traveled (**C**), the foot faults normalized with the distance traveled (**D**), the time immobile (**E**) and the average speed (**F**). The mice were also tested on two different beams. For the notched beam test, the time to cross the beam was measured (**G**) and the number of foot faults were counted (**H**). For the inversed T beam, the time to cross (**I**) and the number of foot faults (**J**) are also illustrated. * indicates a significant difference between C57 and YG8sR, † indicates a significant difference between transgenic mice (Y47R and YG8sR). The results were analyzed by a two-way ANOVA test (Tukey’s multiple comparisons test). * *p* value ˂ 0.05, ** *p* value ˂ 0.005, *** *p* value ˂ 0.0002 and **** ˂ 0.0001. Error bars indicate standard deviation (SD).

**Figure 3 genes-14-01654-f003:**
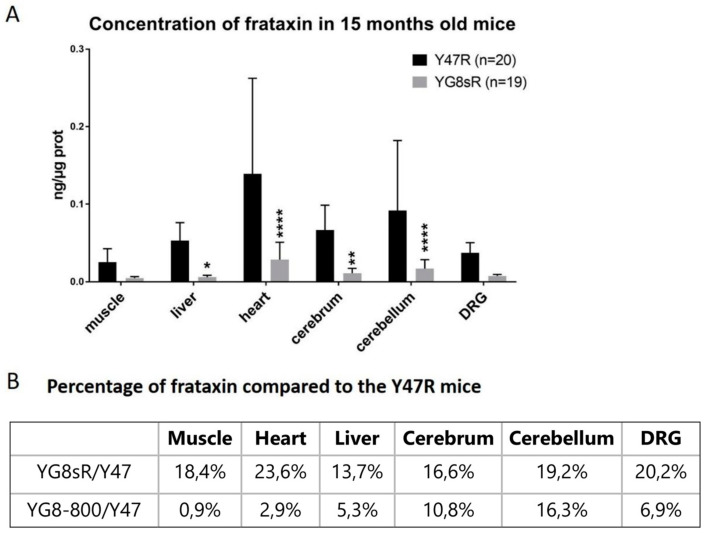
Human frataxin concentration was measured with an ELISA kit in different organs of YG8sR, YG8-800, and Y47R mice. YG8sR compared to Y47R mice (**A**). The frataxin concentrations in YG8sR mice were between 13.7 and 23.6% of those of Y47R mice and were significantly different in the liver, the heart, the cerebrum and the cerebellum. The frataxin concentrations in YG8-800 mice were between 0.9% and 16.3% of those of Y47R mice and were significantly different in the liver, the heart, the cerebrum, the cerebellum and dorsal root ganglions (**B**). The results were analyzed by a two-way ANOVA test (Tukey’s multiple comparisons test). * *p* value ˂ 0.05, ** *p* value ˂ 0.005, and **** ˂ 0.0001. Error bars indicate standard deviation (SD).

**Figure 4 genes-14-01654-f004:**
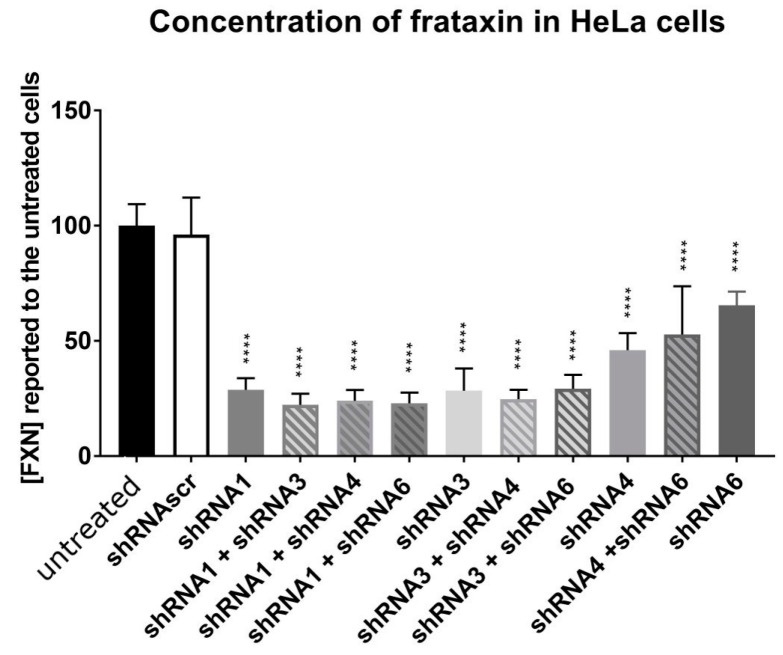
HeLa cells were transfected with plasmids coding for various shRNAs and the frataxin concentration was determined 72 h after transfection. The results were analyzed by a two-way ANOVA test (Tukey’s multiple comparisons test). Control cells were untreated. **** *p* ˂ 0.0001. Error bars indicate standard deviation (SD).

**Figure 5 genes-14-01654-f005:**
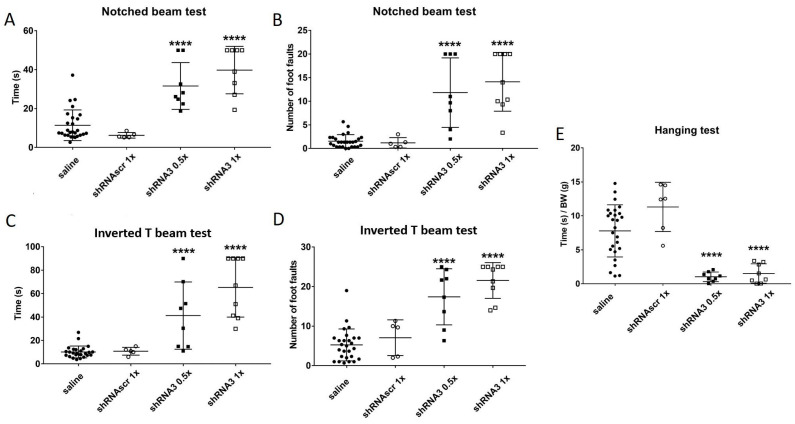
The mice injected with an AAV containing the different shRNAs were monitored with various behavior tests. The time taken by the YG8sR mice to cross the notched beam (**A**) and the inversed T beam (**C**) is illustrated 5 weeks after the injection. The number of foot faults made during the notched beam crossing (**B**) and the inversed T beam crossing (**D**) is also illustrated. In (**E**), the hanging test is quantified as the ratio of hanging time/body weight. The results were analyzed by a two-way ANOVA test (Sidak’s multiple comparisons test). **** *p* value ˂ 0.0001 compared to the saline controls. Error bars indicate standard deviation (SD). The number of animals per group is: saline n = 28, shRNAscr 1 × n = 5, shRNA3 0.5 × n = 8 and shRNA3 1 × n = 9.

**Figure 6 genes-14-01654-f006:**
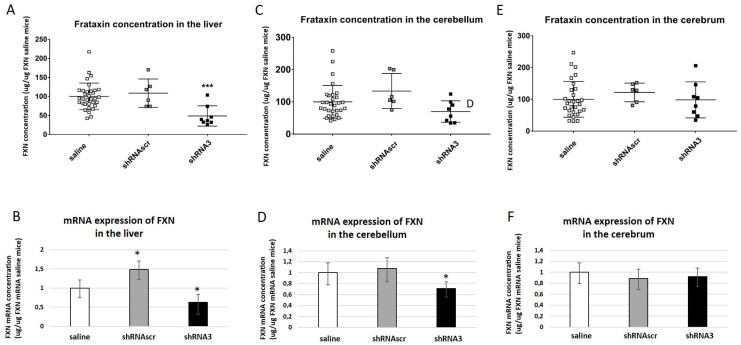
The concentration of frataxin protein was quantified by ELISA (**A**,**C**,**E**) saline n = 35, shRNAscr n = 6, shRNA3 n = 8, and the frataxin mRNA by qRT-PCR (**B**,**D**,**F**) in the liver, the cerebellum, and the cerebrum of mice saline n = 7, shRNAscr n = 6, shRNA3 n = 7. The results of protein concentration were analyzed by a two-way ANOVA test (Tukey’s multiple comparisons test). * *p* value ˂ 0.05, and *** *p* value ˂ 0.0002. Error bars indicate standard deviation (SD). The results from the frataxin mRNA expression were analyzed using the Biorad CFX Maestro software version 2.2.

**Figure 7 genes-14-01654-f007:**
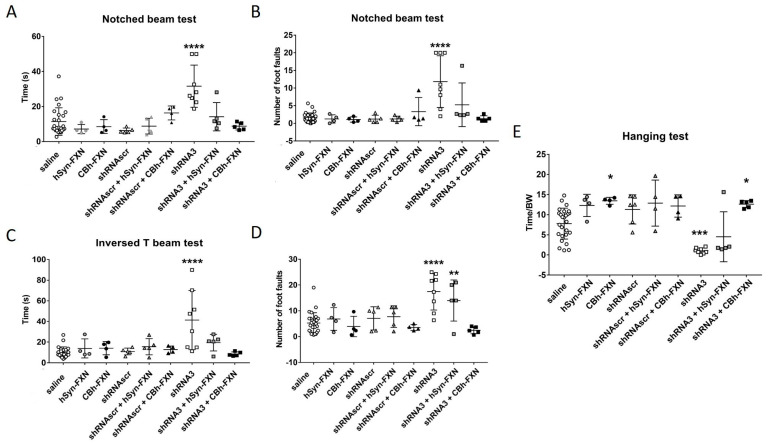
The mice injected with an AAV containing either an shRNA or the frataxin gene or injected with both AAVs were followed with behavior tests. The time taken by the YG8sR mice to cross the notched beam (**A**) and the inversed T beam (**C**) is illustrated 45 days after the injection of AAVs. The number of foot faults made during the notched beam crossing (**B**) and the inversed T beam crossing (**D**) is also illustrated. In (**E**), the results of the hanging test are presented as the ratio of hanging time/body weight. The number of animals per group is saline n = 28, hSyn-FXN n = 4, CBh-FXN n = 4, shRNAscr n = 5, shRNAscr + hSyn-FXN n = 4, shRNAscr + CBh-FXN n = 4, shRNA3 n = 8, shRNA3 + hSyn-FXN n = 5, shRNA3 + CBh-FXN n = 5. The results were analyzed by a two-way ANOVA (Tukey’s multiple comparisons test). * *p* value ˂ 0.05, ** *p* value ˂ 0.005, and **** ˂ 0.0001. Error bars indicate standard deviation (SD).

**Table 1 genes-14-01654-t001:** The concentration of human frataxin was determined and expressed as fold modification relative to the untreated mice for the liver, the cerebellum, and the cerebrum of mice injected with the different AAVs. The control shRNAscr did not significantly modify the frataxin concentration in the 3 tissues. An AAV shRNA3 reduced the expression of frataxin in the liver and cerebellum. The treatment with an AAV coding for frataxin under the CBh promoter increased the expression of frataxin in all three tissues but when the promoter was hSyn the expression was increased in the cerebrum and the cerebellum and only slightly in the liver. The co-injection of AAV-PHP.B coding for the shRNA3 and FXN reduced the expression of FXN compared with AAV-PHP.B coding only for FXN under the same promoter (either CBh or hSyn).

Frataxin Concentration Normalized with Saline Treated Mice						
FXN/FXN saline	shRNAscr	sshRNA3	CBh-FXN	hSyn-FXN	shRNA3 + CBh-FXN	shRNA3 + hSyn-FXN
Liver	1.1 ± 0.4	0.5 ± 0.3	353 ± 245	1.5 ± 0.4	330 ± 238	0.7 ± 0.15
Cerebrum	1.3 ± 0.3	1.0 ± 0.6	134 ± 75	284 ± 197	45 ± 28	164 ± 105
cerebellum	1.4 ± 0.6	0.7 ± 0.3	264 ± 170	113 ± 78	103 ± 56	76 ± 84

## Data Availability

Data is contained within the article or Appendix A.

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
