# Peer review of "Finding an Appropriate Mouse Model to Study the Impact of a Treatment for Friedreich Ataxia on the Behavioral Phenotype"

_genes, 2023, doi:10.3390/genes14081654_

Round 1

Reviewer 1 Report

This manuscript describes two recent murine models of Friedreich Ataxia (FA) that focus on the behavioral phenotype and its potential modification by therapy.  Both are based on earlier models utilizing transgenic expression of a human mutant frataxin on a background of murine frataxin-knockout. The initial model, YG8sR has a single copy of the human mutant transgene with a GAA-repeat of approximately 200.  A newer model, YG8-800, has 800 GAA-repeats.  The authors had been working on a modification of the YG8sR-model prior to the YG8-800 model being developed, so this report describes results with both models.  The goal of these models is to provide a behavioral phenotype that can be quantitatively measured such that improvement by therapy can be evaluated.  The unique model introduced in this paper is based on further knockdown of the frataxin gene on the background of the YG8sR model using AAV-transfection with shRNA directed against that gene.

The results are relatively straightforward and the paper is generally well written.  Both models show some promise in attaining the goals the authors are seeking.  Some of the results, however, are not particularly supportive.  In Figure 2, a number of the tests did not discriminate between wild-type control mice and the models (which included the YG8-800 mice), in particular the hang-time/body weight, the time immobile on the parallel rod floor (except at 5 months), and foot fault on the parallel rod floor.  The only tests that did show significant differences between wild-type controls and the models were the foot faults on the inverted T-beam and the time to cross and the foot faults on the notched beam.  The AAV model tested only at 5 weeks after administration showed greater impairment than even the TG8-800 model although the scatter of data was relatively high.  Is this related to variability in the delivery of the virus to critical elements involved in these behaviors? 

Of note, when interfering shRNAs were introduced simultaneously with knockdown-shRNAs the phenotype was improved to control levels.

One aspect that is somewhat disappointing is the absence of discussion or demonstration of any neuropathological changes associated with any of these models.

There are some minor points of clarification that need to be addressed:

Lines 110 and 112: The definition of the two promoters, CBh and hSyn, should be given.

Line 161: approximately

Line 367:  change “On the opposite” to “In contrast”

Line 370: change “reduced the most the frataxin concentration”  to “reduced frataxin concentration the most”.

Line 400: rod

Line 497: gripped

Line 498: fell

Lines 597-599:  change  “essential to have a good mouse model to also represent the Friedreich ataxia symptoms and detect functional improvements”  to  “essential to have a good mouse model that recreates the symptoms of Friedreich ataxia and allows for detection of functional improvements with treatment”

Line 644: omit the word “testing”

Line 646: change “on” to “of”

Line 651: change “restauration” to “restoration”

Line 654: change “as” to “such as”

Line 655: change “of” to “by”

Line 661:  change “started after a 20 folds increase” to “starting after a 20-fold increase”

Line 662:  change  “a 90 folds frataxin increase”  to “a 90-fold increase of frataxin”

Line 663: insert a comma between “promoters” and “a”

Line 664: omit “Only”

Lines 671-672:  The meaning of this sentence is not clear.  I think you mean that it is important to find a model that is appropriate for what is being tested, e.g., correction of a behavioral phenotype.  I’m not certain what you mean by “biased”.

Minor changes in phrasing and syntax included in the review.

Author Response

All corrections are done with modification tracking in the manuscript.

This manuscript describes two recent murine models of Friedreich Ataxia (FA) that focus on the behavioral phenotype and its potential modification by therapy.  Both are based on earlier models utilizing transgenic expression of a human mutant frataxin on a background of murine frataxin-knockout. The initial model, YG8sR has a single copy of the human mutant transgene with a GAA-repeat of approximately 200.  A newer model, YG8-800, has 800 GAA-repeats.  The authors had been working on a modification of the YG8sR-model prior to the YG8-800 model being developed, so this report describes results with both models.  The goal of these models is to provide a behavioral phenotype that can be quantitatively measured such that improvement by therapy can be evaluated.  The unique model introduced in this paper is based on further knockdown of the frataxin gene on the background of the YG8sR model using AAV-transfection with shRNA directed against that gene.

The results are relatively straightforward and the paper is generally well written.  Both models show some promise in attaining the goals the authors are seeking.  Some of the results, however, are not particularly supportive.  In Figure 2, a number of the tests did not discriminate between wild-type control mice and the models (which included the YG8-800 mice), in particular the hang-time/body weight, the time immobile on the parallel rod floor (except at 5 months), and foot fault on the parallel rod floor.  The only tests that did show significant differences between wild-type controls and the models were the foot faults on the inverted T-beam and the time to cross and the foot faults on the notched beam.  The AAV model tested only at 5 weeks after administration showed greater impairment than even the YG8-800 model although the scatter of data was relatively high.  Is this related to variability in the delivery of the virus to critical elements involved in these behaviors?

Response: We think it is related to the hight variability of delivery.

Of note, when interfering shRNAs were introduced simultaneously with knockdown-shRNAs the phenotype was improved to control levels.

Response: Yes

One aspect that is somewhat disappointing is the absence of discussion or demonstration of any neuropathological changes associated with any of these models.

Response: We tested the movement coordination with behaviour tests and biochemical changes related to frataxin. Do you suggest other neruopathological tests for our next study?

There are some minor points of clarification that need to be addressed:

Lines 110 and 112: The definition of the two promoters, CBh and hSyn, should be given.

Response: added

Line 161: approximately

Response: corrected

Line 367:  change “On the opposite” to “In contrast”

Response: corrected

Line 370: change “reduced the most the frataxin concentration”  to “reduced frataxin concentration the most”.

Response: corrected

Line 400: rod

Response: corrected

Line 497: gripped

Response: corrected

Line 498: fell

Response: corrected

Lines 597-599:  change  “essential to have a good mouse model to also represent the Friedreich ataxia symptoms and detect functional improvements”  to  “essential to have a good mouse model that recreates the symptoms of Friedreich ataxia and allows for detection of functional improvements with treatment”

Response: corrected

Line 644: omit the word “testing”

Response: corrected

Line 646: change “on” to “of”

Response: corrected

Line 651: change “restauration” to “restoration”

Response: corrected

Line 654: change “as” to “such as”

Response: corrected

Line 655: change “of” to “by”

Response: corrected

Line 661:  change “started after a 20 folds increase” to “starting after a 20-fold increase”

Response: corrected

Line 662:  change  “a 90 folds frataxin increase”  to “a 90-fold increase of frataxin”

Response: corrected

Line 663: insert a comma between “promoters” and “a”

Response: corrected

Line 664: omit “Only”

Response: corrected

Lines 671-672:  The meaning of this sentence is not clear.  I think you mean that it is important to find a model that is appropriate for what is being tested, e.g., correction of a behavioral phenotype.  I’m not certain what you mean by “biased”.

Response: corrected

Reviewer 2 Report

Authors provide a manuscript containing three different parts:

1. Behavioral comparison of 2 FRDA mouse models compared to two controls for body weigth, muscle strength and motor deficits followed by FXN protein concentrations of these models in different tissues.

2. Development of shRNAs against FXN and efficacy evaluation of the same in cell culture and in vivo in YG8sR mice using the same behavioral tests as in project part 1.  shRNA levels were confirmed in different tissues and by different methods. FXN protein levels was further measures  in different tissues after treatment with one shRNA

3. FXN was overexpressed in a mouse (not specified) and in parallel expression inhibited by shRNA that was already used in project part 2. Effects were evaluated using behavioral tests and  FXN (probably protein) was quantified in three tissues.

Authors summarize that injecting YG8sR mice with shRNA against FXN increases their phenotype and that YG8-800 mice have a similar phenotype than human FRDA patients.

Title doesn't fit and should be adjusted.

Abstract:

In the beginning, AAV coding shRNA targeting the frataxin gene is mentioned. Is this targeting the human frataxin gene? Is this AAV and the later in the abstract mentioned AAV-PHP.B the same?

Introduction:

Some phrasings are strange and should be checked.

·       Our group has previously used these mouse models... (lane 63)

·       Evolution of the disease (lane 64)

·       phenotype closer to that to the Friedreich ataxia (lane 79)

·       genes are knockout but the model mouse contains (lane 87)

·       The co-injection of AAV-PHP.B coding for a shRNA and for the frataxin gene as a treatment (lane 100)

Listing of all available FRDA mouse models is nice, but in general not necessary. Either shorten this chapter or even provide a table for comparison of models.

The end of the introduction contains too many details about used methods.

Material and Methods:

2.1.: This chapter is completely confusing and should be better structured. Some wording is lab slang (oligos) but not appropriate for a manuscript.

At the beginning and the very end of the chapter, authors talk about a vector, in between only about plasmids. Is this on purpose? What is the difference here?

Provide details about PHP.B serotype. Where does it drive expression?

2.3. please rephrase “normal mouse” to control mouse or similar.

How many animals were used per group? Age? Sex? Housing conditions?

2.4.

·       “specific nervous delivery” the delivery is nervous? I have doubts about that, please rephrase.

·       What is a “general delivery”? ubiquitously?

·       Here is suddenly explained what AAV stands for, this belongs at the beginning of the methods.

·       2.4. and 2.5. belong in front of 2.3.

·       Information provided in 2.4. needs to be mentioned in 2.1. to make it better understandable (e.g., what all these promoters are good for)

2.5. (3x1011genome copies) what are these genome copies?

Also here, this is completely confusing. Were mice injected with shRNA or with AAV vectors that contained such shRNA. Please always use the same expressions.

Names of shRNA should contain the information that there are against FXN.

Age of mice at different stages of study should be mentioned more specifically. At which age were animals tested in the different behavioral tests? A Time schedule would be helpful.

Sacrification of mice should be described in own chapter. Which euthanasia method was used? How was tissue stored?

2.6.1. This test is not commonly known, please explain set up. Providing a reference is not enough. It is mentioned that foot faults were measured by apparatus and later by ANY-maze software. What is correct?

2.6.3. “the average number was…”(lane 190) number of what? How was recording done?

2.6.4. “the average number was…”(lane 196) number of what? How was recording done? Why is this test called “inverted T beam”. The provided explanation sounds like a standard beam walk test. Why was a Plexiglas beam used instead of a wooden beam?

2.7. fluorescent signal was normalized to what? Exposure time should be always the same to be able to compare animals / groups!?

2.8.: “detection of the AAV..” (lane 218) did you detect sequence of AAV or shRNA/FXN? Suppl. Table 1 does not provide AAV primer.

2.10: “molecular biology water” what is that? Provide more details about the whole procedure. With the currently provided information, a repetition of experiments would not be possible.

2.11. this chapter describes protein quantification. When did you describe the protein extraction process? Protein quantification needs to be described in more detail.

2.12.: what is meant with “average±SD”? is average = mean? Please explain SD. Please describe statistical tests also here, not only in figure legends. Is this **p < 0.003, ***p < 0.0003 correct? This would be very uncommon. Usually the following is used: *p < 0.05, **p < 0.01, ***p < 0.001, and ****p <0.0001.

3.1.1.: there is way too much methodology described in the beginning. Age of mice, group size, measurement of body weight, all these belong in methods section.

Figure 2: y-axes: unit of measurement should always be provided in parentheses as done in A and C.      B, D, E, F, H, J needs to be corrected.

Error bars need to be explained. What are they showing SD or SEM? Please also correct in all other figures.

Lane 299: Information about asterisks and further symbols doesn’t fit to symbols used in figure.

Lane 301: p value levels are again unusual and should be kept the same throughout the manuscript.

Lane 303: Information belongs in methods not the results.

Lane 306: “The distance is inferior for..” makes no sense

Lane 308: “C57 healthy model…” is slang. Please always use the same expression for these mice.

Lane 322: increase in what? Which test?

Lane 328: this summary is not specific enough. A mild phenotype in what?

The whole description of behavioral results needs to be written more precisely and phrasing needs to be corrected.

3.1.2.

Lane 332: Are you talking about RNA or protein here?

Fig.3A: Why are YG8-800 mice not shown in A? Legend of A and B doesn’t fit to figure, seems to be switched? Why are error bars so strong? When looking at y-axis in A, differences between groups seem only minor, as their range between 0 and 0.2, specifically when considering the high deviations. This should be mentioned.

Unify p-values between figures.

Lane 352: Why “later on, the YG8-800 mouse model was available by JAX”. The model was included in the here presented manuscript from figure 2 onwards. This information is thus just confusing. Remove.

3.2.1:

Lane 377: Information is not correct. Some combinations reduced FXN concentrations stronger that any single shRNA. The difference was probably not significant?

Was protein or RNA analyzed here?

Lane 382: compared to which control cells? Labeling of first bar (black) is confusing. Add information about treatment. If cells were not treated, add “untreated”.

Lane 387: Base on information provided in lane 363, shRNAs were already inserted in AAV-PHP.B vector. Were the same constructs used?

3.2.2.1: Please rephrase title.

Lane 390: increase phenotype of what? How/where did you administer the shRNA? shRNA against what? Please be more precise!

Figure 5C and D and S2C and D: Graph title: inversed or inverted?

Fig 5 and S2: number of animals per group doesn’t belong into x-axis. Add information to legend.

3.2.2.2

Lane 428: Cr and Ce are not mentioned and explained.

Figure 6: Measurement of mCherry signal, AAV and mCherry gDNA quantification provides only indirect evidence for shRNA levels. Data should be moved to Supplementary data.

3.2.3

Figure 7: y-axes of all graphs miss a unit.

3.2.4:

Lane 482: if sequences are almost identical, did you test for effect of shRNA on human vs. murine FXN protein levels?

Chapter needs more explanation why these tests were performed. It needs to be mentioned that the used AAV serotype was not neuro-specific and selected promotors were thus chosen to selectively express FXN in neuronal/non-neuronal tissues.

Until this point, I understood that AAV expression using different promoters was used to also express shRNA but it seems that they were used to express FXN? Is this right? Based on Methods, this is not recognizable. Authors need to separate methods about shRNA and FXN expression more clearly. Currently, this is very confusing. Authors should provide information in separate chapters. Was human or mouse FXN expressed?

3.2.5 The availability of YG8-800 mice should not be mentioned. It is irrelevant for the provided manuscript.

This chapter does not belong here. As data of different figures are compared with each other, this chapter belongs in the discussion. Again, very confusing.

Discussion

First chapter: It seems that authors in the beginning only talk about already published data. Is this correct? If yes, authors need to say so more clearly. Authors need to better explain what are the similarities and differences between already published and own data.

Lane 586: Reference 41: Is this study relevant for data presented here? It seems to be about ALDH2 and the kidney, not FXN.

Lane 591: It is not interesting for the reader how the shRNA was named.

Reference 42: It is sufficient to mention the conclusion of this reference, details are not needed as they are more confusing than helping.

Lane 607: mouse model for what? Studies of reference 27 need to be better explained. It currently is not clear what doxycycline does in this mouse. Is it inducing or inhibiting FXN?

Lane 616 and following: This is illogical. Age groups don’t fit to your conclusion.

Lane 624: reference 27, what is your conclusion out of this study for treatments?

Lane 632: information about AAV-PHP.B specificity needs to be mentioned in methods.

Lane 635: AAV-PHP.B-shRNA -this shRNA is against what?

Lane 639: Study of reference 16 needs to be better explained. Why is paralbumine relevant for FRDA?

Lane 637-655: how is this all relevant to the here presented data?

Lane 667: Wording, not the CBh promoter caused changes but FXN expressed under control of this promoter. Be precise!

Lane 693: the information about gene therapy should be discussed in the appropriate section. Here, it comes out of the blue.

In Summary: Study design and results are good but authors need to be more precise about their wording and better guide reader through the experiments and conclusions.

Most parts of the manuscript need language editing. In some parts, language problems cause problems to understand the context. Language of Discussion is better.

Author Response

All corrections are done with modification tracking in the manuscript.

Authors provide a manuscript containing three different parts:

    Behavioral comparison of 2 FRDA mouse models compared to two controls for body weigth, muscle strength and motor deficits followed by FXN protein concentrations of these models in different tissues.

    Development of shRNAs against FXN and efficacy evaluation of the same in cell culture and in vivo in YG8sR mice using the same behavioral tests as in project part 1.  shRNA levels were confirmed in different tissues and by different methods. FXN protein levels was further measures  in different tissues after treatment with one shRNA

    FXN was overexpressed in a mouse (not specified) and in parallel expression inhibited by shRNA that was already used in project part 2. Effects were evaluated using behavioral tests and  FXN (probably protein) was quantified in three tissues.

Authors summarize that injecting YG8sR mice with shRNA against FXN increases their phenotype and that YG8-800 mice have a similar phenotype than human FRDA patients.

Title doesn't fit and should be adjusted.

Response: corrected

Abstract:

In the beginning, AAV coding shRNA targeting the frataxin gene is mentioned. Is this targeting the human frataxin gene? Is this AAV and the later in the abstract mentioned AAV-PHP.B the same?

Response: yes, it targets the human FXN gene and is the same AAV. PHP.B

Introduction:

Some phrasings are strange and should be checked.

    Our group has previously used these mouse models... (lane 63)

    Evolution of the disease (lane 64)

    phenotype closer to that to the Friedreich ataxia (lane 79)

    genes are knockout but the model mouse contains (lane 87)

    The co-injection of AAV-PHP.B coding for a shRNA and for the frataxin gene as a treatment (lane 100)

Response: corrected

Listing of all available FRDA mouse models is nice, but in general not necessary. Either shorten this chapter or even provide a table for comparison of models.

Response: (I forgot to add it to the manuscript, but here is a comparative table of mouse models. The editor will help add this table on the manuscript.)

Mouse model

Genotype

Phenotype

αMyhc (myosin heavy chain)

Cre recombinase conditionnal knockdown (excision of exon 4) of mouse frataxin under a cardiac promoter (αMyhc).

Decreased cardiac frataxin and cardiac performance under stress.

MCK-Cre (muscle creatine kinase)

Cre recombinase conditionnal knockdown of frataxin under a muscle creatine kinase promoter.

Severe, with cardiac involvement (lifespan 90 days).

NSE-cre (neuron-specific enolase)

Cre recombinase conditionnal knockdown of frataxin under a neuro-specific enolase promoter.

Severe, with cardiac involvement (lifespan 30 days).

Pvalbtm1(Cre)Arbr/J

Conditional knockdown of frataxin under a parvalbumin promoter (specific to proprioceptive neurons).

Coordination impairment.

KIKO (Fxn(tm1Mkn/J)

Expansion of 230 GAA in one of the mouse Fxn gene and a deletion of the exon 4 in the other Fxn gene.

Severe coordination impairment.

FRDAkd

Inducible and reversible condition: transgene containing a shRNA against mouse frataxin under a doxycycline-inducible H1 promotor.

Significant frataxin protein reduction with decreased locomotor activity and coordination.

YG8R

Mouse FXN gene knockout + human frataxin transgene from patient containing two GAA expansions (90 + 190).

Mild frataxin protein reduction and slight coordination impairment.

YG8sR

Mouse FXN gene knockout + human frataxin transgene from patient containing one GAA expansion (250-300).

Medium frataxin protein reduction and slight coordination impairment.

YG8-800

Mouse FXN gene knockout + human frataxin transgene from patient containing one GAA expansion (800).

Increased frataxin protein reduction and coordination impairment comparable to human + cardiac hypertrophy.

The end of the introduction contains too many details about used methods.

Response: corrected

Material and Methods:

2.1.: This chapter is completely confusing and should be better structured. Some wording is lab slang (oligos) but not appropriate for a manuscript.

Response: corrected

At the beginning and the very end of the chapter, authors talk about a vector, in between only about plasmids. Is this on purpose? What is the difference here?

Response: corrected

Provide details about PHP.B serotype. Where does it drive expression?

Response: added

2.3. please rephrase “normal mouse” to control mouse or similar.

Response: corrected

How many animals were used per group? Age? Sex? Housing conditions?

Response: added

2.4.

“specific nervous delivery” the delivery is nervous? I have doubts about that, please rephrase.

Response: corrected

What is a “general delivery”? ubiquitously?

Response: yes

Here is suddenly explained what AAV stands for, this belongs at the beginning of the methods.

Response: corrected

2.4. and 2.5. belong in front of 2.3.

Response: corrected

    Information provided in 2.4. needs to be mentioned in 2.1. to make it better understandable (e.g., what all these promoters are good for)

Response: corrected

2.5. (3x1011genome copies) what are these genome copies?

Response: viral units

Also here, this is completely confusing. Were mice injected with shRNA or with AAV vectors that contained such shRNA. Please always use the same expressions.

Response: an AAV containing the shRNA sequence in a plasmid

Names of shRNA should contain the information that there are against FXN.

Response: corrected

Age of mice at different stages of study should be mentioned more specifically. At which age were animals tested in the different behavioral tests? A Time schedule would be helpful.

Response: added

Sacrification of mice should be described in own chapter. Which euthanasia method was used? How was tissue stored?

Response: corrected

2.6.1. This test is not commonly known, please explain set up. Providing a reference is not enough. It is mentioned that foot faults were measured by apparatus and later by ANY-maze software. What is correct?

Response: added

2.6.3. “the average number was…”(lane 190) number of what? How was recording done?

Response: corrected

2.6.4. “the average number was…”(lane 196) number of what? How was recording done? Why is this test called “inverted T beam”. The provided explanation sounds like a standard beam walk test. Why was a Plexiglas beam used instead of a wooden beam?

Response: We used a plexiglass beam made from a supplier for animal behaviour tests in our hospital.

2.7. fluorescent signal was normalized to what? Exposure time should be always the same to be able to compare animals / groups!?

Response: We do not have the samples anymore but we will do so next time.

2.8.: “detection of the AAV..” (lane 218) did you detect sequence of AAV or shRNA/FXN? Suppl. Table 1 does not provide AAV primer.

Response: corrected in text (we detected the sequence of FXN and mCherry in the shRNA plasmid)

2.10: “molecular biology water” what is that? Provide more details about the whole procedure. With the currently provided information, a repetition of experiments would not be possible.

Response: It is bottled water, which is RNAse and DNAse free.

2.11. this chapter describes protein quantification. When did you describe the protein extraction process? Protein quantification needs to be described in more detail.

Response: corrected

2.12.: what is meant with “average±SD”? is average = mean? Please explain SD. Please describe statistical tests also here, not only in figure legends. Is this **p < 0.003, ***p < 0.0003 correct? This would be very uncommon. Usually the following is used: *p < 0.05, **p < 0.01, ***p < 0.001, and ****p <0.0001.

Response: yes, we meant mean instead of average.

3.1.1.: there is way too much methodology described in the beginning. Age of mice, group size, measurement of body weight, all these belong in methods section.

Response: corrected

Figure 2: y-axes: unit of measurement should always be provided in parentheses as done in A and C.      B, D, E, F, H, J needs to be corrected.

Response: corrected

Error bars need to be explained. What are they showing SD or SEM? Please also correct in all other figures.

Response: SD

Lane 299: Information about asterisks and further symbols doesn’t fit to symbols used in figure.

Response: corrected

Lane 301: p value levels are again unusual and should be kept the same throughout the manuscript.

Response: corrected

Lane 303: Information belongs in methods not the results.

Response: corrected

Lane 306: “The distance is inferior for..” makes no sense

Response: corrected

Lane 308: “C57 healthy model…” is slang. Please always use the same expression for these mice.

Response: corrected

Lane 322: increase in what? Which test?

Response: foot faults

Lane 328: this summary is not specific enough. A mild phenotype in what?

Response: corrected, movement coordination impairment

The whole description of behavioral results needs to be written more precisely and phrasing needs to be corrected.

Response: corrected

3.1.2.

Lane 332: Are you talking about RNA or protein here?

Response: protein

Fig.3A: Why are YG8-800 mice not shown in A? Legend of A and B doesn’t fit to figure, seems to be switched? Why are error bars so strong? When looking at y-axis in A, differences between groups seem only minor, as their range between 0 and 0.2, specifically when considering the high deviations. This should be mentioned.

Response: 800 is not shown because it is not visible compared to the error bars of the other groups. Therefore, we made a table.

Unify p-values between figures.

Response: corrected

Lane 352: Why “later on, the YG8-800 mouse model was available by JAX”. The model was included in the here presented manuscript from figure 2 onwards. This information is thus just confusing. Remove.

Response: corrected

3.2.1:

Lane 377: Information is not correct. Some combinations reduced FXN concentrations stronger that any single shRNA. The difference was probably not significant?

Response: not significant

Was protein or RNA analyzed here?

Response: protein

Lane 382: compared to which control cells? Labeling of first bar (black) is confusing. Add information about treatment. If cells were not treated, add “untreated”.

Response: untreated

Lane 387: Base on information provided in lane 363, shRNAs were already inserted in AAV-PHP.B vector. Were the same constructs used?

Response: yes

3.2.2.1: Please rephrase title.

Response: corrected

Lane 390: increase phenotype of what? How/where did you administer the shRNA? shRNA against what? Please be more precise!

Response: added

Figure 5C and D and S2C and D: Graph title: inversed or inverted?

Response: inverted (corrected)

Fig 5 and S2: number of animals per group doesn’t belong into x-axis. Add information to legend.

Response: corrected

3.2.2.2

Lane 428: Cr and Ce are not mentioned and explained.

Response: corrected

Figure 6: Measurement of mCherry signal, AAV and mCherry gDNA quantification provides only indirect evidence for shRNA levels. Data should be moved to Supplementary data.

Response: corrected

3.2.3

Figure 7: y-axes of all graphs miss a unit.

Response: corrected

3.2.4:

Lane 482: if sequences are almost identical, did you test for effect of shRNA on human vs. murine FXN protein levels?

Response: not yet

Chapter needs more explanation why these tests were performed. It needs to be mentioned that the used AAV serotype was not neuro-specific and selected promotors were thus chosen to selectively express FXN in neuronal/non-neuronal tissues.

Response: corrected

Until this point, I understood that AAV expression using different promoters was used to also express shRNA but it seems that they were used to express FXN? Is this right? Based on Methods, this is not recognizable. Authors need to separate methods about shRNA and FXN expression more clearly. Currently, this is very confusing. Authors should provide information in separate chapters. Was human or mouse FXN expressed?

Response: human

3.2.5 The availability of YG8-800 mice should not be mentioned. It is irrelevant for the provided manuscript.

Response: corrected

This chapter does not belong here. As data of different figures are compared with each other, this chapter belongs in the discussion. Again, very confusing.

Response: corrected

Discussion

First chapter: It seems that authors in the beginning only talk about already published data. Is this correct? If yes, authors need to say so more clearly. Authors need to better explain what are the similarities and differences between already published and own data.yes

Response: corrected

Lane 586: Reference 41: Is this study relevant for data presented here? It seems to be about ALDH2 and the kidney, not FXN.relevance of promoter specificity (added to text)

Response: yes, relevance of promoter specificity.

Lane 591: It is not interesting for the reader how the shRNA was named.

Response: corrected

Reference 42: It is sufficient to mention the conclusion of this reference, details are not needed as they are more confusing than helping.

Response: corrected

Lane 607: mouse model for what? Studies of reference 27 need to be better explained. It currently is not clear what doxycycline does in this mouse. Is it inducing or inhibiting FXN?

Response: inhibiting FXN

Lane 616 and following: This is illogical. Age groups don’t fit to your conclusion.

Response: corrected

Lane 624: reference 27, what is your conclusion out of this study for treatments?

Response: added

Lane 632: information about AAV-PHP.B specificity needs to be mentioned in methods.

Response: corrected

Lane 635: AAV-PHP.B-shRNA -this shRNA is against what?

Response: frataxin

Lane 639: Study of reference 16 needs to be better explained. Why is paralbumine relevant for FRDA?

Response: Promoter for proprioceptive neurons

Lane 637-655: how is this all relevant to the here presented data?

Response: corrected

Lane 667: Wording, not the CBh promoter caused changes but FXN expressed under control of this promoter. Be precise!

Response: corrected

Lane 693: the information about gene therapy should be discussed in the appropriate section. Here, it comes out of the blue.

Response: corrected

In Summary: Study design and results are good but authors need to be more precise about their wording and better guide reader through the experiments and conclusions.

Response: corrected